# Mask-Wearing during the COVID-19 Pandemic: A Theoretical Analysis from the Perspective of Public Health Ethics

**Akira Akabayashi** [1,2,*] , **Aru Akabayashi** [1] **and Eisuke Nakazawa** [1]

1   Department of Biomedical Ethics, University of Tokyo Faculty of Medicine, 7-3-1 Hongo, Bunkyo-ku, Tokyo 113-0033, Japan
2   Division of Medical Ethics, New York University School of Medicine, 227 East 30th Street, New York, NY 10016, USA
*   Correspondence: akira.akabayashi@gmail.com or akirasan-tky@umin.ac.jp; Fax: +81-35841-3319

**Abstract:** Do we have the right to wear masks during an infectious disease pandemic? If so, what is the underlying philosophical justification for this? During the COVID-19 pandemic, most people wore masks. Should the government be able to intervene to enforce mask wearing? It should be noted that the government's encouragement to wear masks does not mean that people are encouraged to ignore them. In the field of public health ethics, many current debates boil down to establishing a balance between "individual freedom" and "the public good". However, a clear definition of "the public good" has yet to emerge, which can make this debate difficult. Based on our philosophical analysis, we propose the following as a new right in the field of public health ethics: the "right to mask for self-protection". Based on our proposed "right to mask for self-protection", we offer a justification for the argument that all people have the right to wear a mask during an infectious disease pandemic or endemic.

**Keywords:** hygiene mask; medical mask; dust mask; self-protection; natural law; harm principle; public health ethics





## 1. Introduction

The global pandemic of COVID-19 continues to impact humans worldwide, with little reprieve in sight. On Thursday 18 August 2022, we conducted brief surveys of shoppers at major supermarkets in suburban areas of Tokyo, Japan and Rochester, NY, USA. We began surveying shoppers at 16:00 (during the peak shopping time), from one observation point at each site. We surveyed approximately 206 customers passing through the observation point and counted the number of people wearing masks. In the U.S., 34 (16.5%) wore masks, while in Japan, all 206 (100%) wore masks. What does this difference indicate?

The stereotypical explanation, i.e., that Japan is a collectivist society while America is individualist, is far too overused and overly schematic. Despite the fact that public health ethics [1] was a main focal point of the discussion during the pandemic, we have yet to come up with a clear solution with regard to balancing "individual freedom" and "the public good". This study focuses on mask wearing as a public health response to the COVID-19 pandemic and preliminarily expands the theoretical discussion from the standpoint of public health ethics.

## 2. The History of Hygiene Masks

Let us take a brief look back at the history of hygiene masks. The dust mask, which is well known to many, is worn to prevent the inhalation of airborne particulates. The medical or surgical mask, which is also familiar to most, is designed to prevent infection, mainly from airborne droplets. The unique mask worn by doctors in 17th century Europe to combat the Black Plague (glandular plague) is also fairly well known (Figure 1). This plague was a systemic invasive infection caused by *Yersinia pestis*, which spreads from

person to person via a droplet infection through airway secretions discharged from a patient with pneumonic plague. Accordingly, masks worn by doctors during this plague were categorized as medical masks. In Japan, medical masks became widely used after the Spanish flu (commonly known as the H1N1 subtype of influenza that spread worldwide from 1918 to 1920). Thus, masks have historically been used for infection prevention.

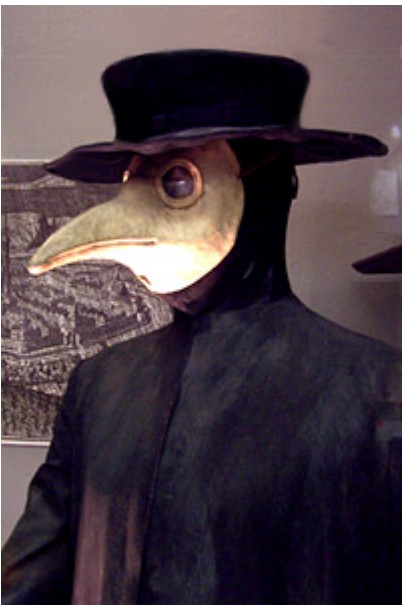

**Figure 1.** Special physician clothes for preventing pestilence (Germany, XVII century) at Jena. Juan Antonio Ruiz Rivas-Enciclopedia Libre en Español.

### 3. Discussion

*3.1. Masks during an Infectious Disease Pandemic or Endemic*

The following is a statement issued by the WHO during the 2009 Influenza A (H1N1) outbreak [2]:

*If masks are worn, proper use and disposal is essential to ensure they are potentially effective and to **avoid any increase in risk of transmission** associated with the incorrect use of masks. The following information on correct use of masks derives from the practices in health-care settings.*

- *Place mask carefully to cover mouth and nose and tie securely to minimise any gaps between the face and the mask;*
- *While in use, avoid touching the mask—whenever you touch a used mask, for example, when removing or washing, clean hands by washing with soap and water or using an alcohol-based handrub;*
- *Replace masks with a new clean, dry mask as soon as they become damp/humid;*
- *Do not re-use single-use masks—discard single-use masks after each use and dispose of them immediately upon removing.*

*Although some alternative barriers to standard medical masks are frequently used (e.g., cloth mask, scarf, paper masks, rags tied over the nose and mouth), there is insufficient information available on their effectiveness. If such alternative barriers are used, they should only be used once or, in the case of cloth masks, should be cleaned thoroughly between each use (i.e., wash with normal household detergent at normal temperature). **They should be removed immediately after caring for the ill.** Hands should be washed immediately after removal of the mask.*

In other words, of the different varieties of sanitary masks, dust masks are used to protect oneself, while medical masks have **two** functions, namely to "protect oneself" and to "prevent the spread of infection".

There are also other problematic issues [3,4], and the problem of mask wearing (or not) has become fairly politicized in the U.S. These are prime examples of why trying to

weigh "individual freedom" against "public good" does not work well. Why have some Americans refused to wear masks, even when public health experts strongly recommended doing so? [5,6] Is it because Americans are individualistic, and their individual freedoms are allowed to override the "public good"? As stated above, the concept of "public good" is vague, and may only indicate that people should wear masks in order to prevent the spread of infection to society. However, this is just one of the two functions of medical masks listed above. These discussions lead us to ask the following question: "What is the origin of the right to wear a mask for self-protection, which is one of the other important functions of medical masks?". On which theory or principle is the right to wear a mask for self-protection based?

*3.2. Philosophical Debates on Mask Wearing*

In terms of preventing the spread of infection, Mill's "harm principle" is highly relevant from a public health ethics perspective. According to this principle, as a general rule, only acts that cause harm to others can be prohibited by the government or public opinion [7]. Thus, it is permissible for the government to recommend that people wear masks during a pandemic, because not wearing a mask during that time will cause harm to others.

A consideration of Mill's harm principle is necessary when discussing the "right of (to) stupidity". This right guarantees the freedom to commit a stupid act, even if the act is evaluated or judged as stupid by others, as long as the act does not cause harm to others, without interference from anyone as far as the person is concerned. In a state of endemicity, not wearing a mask can also be justified by this right of (to) stupidity.

*3.3. On Contemporary Natural Rights and the Right to Live or Life*
3.3.1. Mask Wearing during a Pandemic

The modern right to live or life is thought to be attributed to, and dates back to, Hobbes, and it is a concept derived from the idea of natural law [8]. The concepts of natural rights and natural law originated in ancient Greece and can be thought of as rights that each human being possesses at birth.

John Locke set forth four rights that are violable by no one; these included the rights to live or life, health, liberty, and possessions [9]. Within self-preservation, he included a broader concept of liberty and the right to own property. Locke's ideas gave theoretical legitimacy to capitalism and civil society and greatly influenced civil revolutions such as the American Revolution.

In fact, in many of today's democracies, rights that have been regarded as natural rights are stipulated in constitutions. In Japan's Constitution, for example, natural rights are guaranteed as permanent rights under the guise of fundamental human rights.

Thus, when we look at the history of natural rights, we see that every human being born into this world possesses the right to self-preservation, i.e., to protect his or her own life. In the case of the COVID-19 pandemic, each of us has the right to wear a mask in order to preserve our own life. Here, we tentatively and preliminarily call this the "right to mask for self-protection" in the public health context. A deeper philosophical discussion on this matter is beyond the scope of this short communication and will be conducted elsewhere, since this brief piece is intended for public health (especially health protection and disease prevention) and other disciplines, not for philosophers.

Briefly, the "right to mask for self-protection" is the innate right of human beings (based on natural law) to protect their own lives to the extent that they do not harm others (it would not be upheld, for example, if they had to kill 10 people in order to live). In other words, one has the right to defend oneself by any means necessary to protect one's own life, so long as others are not harmed.

The "right to mask for self-protection" is a concept deeply dependent upon natural law theory and Mill's harm principle. The "right to live or life" originally placed emphasis on one's right not to be killed by others (i.e., by police or the government). Indeed, the "right to

mask for self-protection" could be considered a specific right within the broader collection of the rights to live or life. However, our proposed "right to mask for self-protection" applies primarily to public health.

We argue, therefore, that during the COVID-19 pandemic, in the absence of effective treatments or vaccines, people have the right to wear masks and defend themselves. This "right to mask for self-protection" perspective may impact the global response to an infectious disease pandemic. The essential role of the medical mask is "self-protection". Thus, we surmise that if the discussion in the U.S. had brought up the "right to mask for self-protection" against "individual freedom", the discussion might have taken a completely different course. Moreover, we consider it unethical to interfere with people who invoke the "right to mask for self-protection", unless there is a substantial basis to override it.

One key question is whether government intervention is permissible or not. A government has a strong responsibility to protect the lives of the people it serves. It might be ethically justifiable for a government to intervene to protect the lives of citizens (so that citizens can invoke their "right to wear a mask for self-protection"), such as by requiring citizens to wear masks when neither a vaccine nor medical treatment is available. However, there was controversy and debate, with some opponents of government protection policies arguing for a "right to mask for self-protection" when they were actually against wearing masks themselves. It should also be noted that the government's encouragement to wear masks does not mean that people are encouraged to ignore them.

### 3.3.2. Mask Wearing during an Endemic

Another question is "[D]o people have the "right to mask for self-protection" during an endemic?". We would argue that, even in an endemic, people have the "right to mask for self-protection". For example, why did a small number of people in the supermarket in Rochester wear masks? They may have been invoking the "right to mask for self-protection" because the risk of infection is not yet **zero**. Therefore, we argue that even in the endemic stages of a pandemic, all people have the "right to mask for self-protection".

There may also be such a thing as the "right NOT to mask" [10,11]. We stated earlier that the "right of (to) stupidity" might offer some justification for those who do not wear masks during pandemics. However, the reality is that in Japan, the right not to mask is being threatened. Nearly 100% of the population wears masks during endemic times. This may be due to ongoing government campaigns, which have influenced the public and imprinted the idea that those who do not wear masks are negatively impacting others who may contract the virus. This may also be explained, in part, by the overused notion of collectivism, i.e., that the Japanese must conform to norms followed by others in society.

Another issue related to masks is severe acute respiratory syndrome coronavirus 2 (SARS-CoV-2). Although it is clear that SARS-CoV-2 is transmitted through human respiratory droplets and direct contact, the potential for aerosol transmission is poorly understood [12,13]. Liu et al. indicate that room ventilation, open space, the sanitization of protective apparel, and the proper use and disinfection of toilet areas can effectively limit the concentration of SARS-CoV-2 RNA in aerosols [14]. Ueki et al. found that airborne simulation experiments showed that cotton masks, surgical masks, and N95 masks provide some protection from the transmission of infective SARS-CoV-2 droplets/aerosols; however, medical masks (surgical masks and even N95 masks) could not completely block the transmission of virus droplets/aerosols, even when sealed [15].

### 4. Conclusions

Every individual has the "right to mask for self-protection" during a pandemic, as well as during its endemic stage. Wearing a mask does not cause any inconvenience to others. We believe that the "right to mask for self-protection" is something that cannot be overridden by the right to individual freedom. Although this study only provides a preliminary discussion, we hope to construct a more solid philosophical foundation for the "right to mask for self-protection" in the context of public health.

We anticipate that the introduction of the "right to mask for self-protection" perspective will contribute to a new insight, which will enhance the debate on public health ethics beyond the impasse of simply prioritizing between individual freedom and the ambiguous public good.

**Author Contributions:** Conceptualization, A.A. (Aru Akabayashi); methodology, A.A. (Aru Akabayashi) and E.N.; writing—original draft preparation, A.A. (Akira Akabayashi); writing—review and editing, A.A. (Aru Akabayashi), E.N., and A.A. (Akira Akabayashi); project administration, A.A. (Akira Akabayashi). All authors have read and agreed to the published version of the manuscript.

**Funding:** This research received no external funding.

**Institutional Review Board Statement:** Not applicable.

**Informed Consent Statement:** Not applicable.

**Data Availability Statement:** Not applicable. The image used in the manuscript is CC-BY-SA, so there are no copyright issues.

**Conflicts of Interest:** The authors declare no conflict of interest.

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
