# Peer review of "Mask-Wearing during the COVID-19 Pandemic: A Theoretical Analysis from the Perspective of Public Health Ethics"

_2673-8430, doi:10.3390/biomed2040030_

Round 1

Reviewer 1 Report

Thank you for submitting this topical and interesting paper. 

It is clearly written (some minor edits needed) and it addresses something that remains highly topical. I don't have any major disagreements with your conclusions. However I do find the question you focus on rather perplexing, why ask is there a 'right to wear a mask'? rather than the more challenging 'is there a duty to wear a mask'?

I think that most people will be non-plussed by the idea that there is a right to wear a mask since, at one end of the scale people generally have  a right to wear what they want, at least for Millian reasons of non-interference. There are of course a few instances when this has been challenged (e.g. the veil or hijab etc). You don't actually say very much, empirically, about the specific use of surgical face-masks in the context of infectious diseases, evidence that the wearing of masks does convey significant benefit would strengthen your claim that it is based on a 'right to life'.

Having made the claim that there is such a right you do not go on to develop the more interesting implication that there is a duty for the state to provide masks for people (after all rights are always twinned with duties). It seems to me that without this acknowledgement mask-wearing is just a personal preference rather than a substantive right. 

There are two other areas that remain under-developed. One is your claim at 157-160 the right not to mask is being threatened in Japan. This needs some further discussion and in particular the claim would benefit from some account of the importance of this 'freedom' especially as you suggest that mask wearing is both self-protecting and protecting of others, thus engaging the harm principle.

The second area is linked and relates to the culture of individualism you note is a particular feature of the USA. I am curious as to what you make of this culture in terms of the implied (robust) right to non-interference. Is this a right that trumps others, is it a right to 'stupidity' or what?

I think that you need to do a little more to show that the right to wear a mask is a substantive right (right to life) and not merely a matter of individual preference. However I still think that the more interesting question is whether there is a duty to wear a mask.

Author Response

Rev 1 Comments and Suggestions for Authors

Thank you for submitting this topical and interesting paper.

It is clearly written (some minor edits needed) and it addresses something that remains highly topical. However I do find the question you focus on rather perplexing, why ask is there a 'right to wear a mask'? Rather than the more challenging 'is there a duty to wear a mask'?

The COVID-19 pandemic is now finally about to subside. Our issue is not the wearing of masks in a pandemic situation, but the wearing of masks in a situation where the pandemic is subsiding. In fact, the background of our discussion is the sad situation that we are facing, especially in Europe and the United States, where there are cases of people being accused of wearing masks simply because they are wearing masks (see manuscript references 3 and 4.) Although it is too sensitive a matter to be written in an explicit way, our paper was written with the intention of alleviating prejudice against the Orient, the friction between cultures, and the life-threatening discrimination against African-Americans. Therefore, we must now insist that it is OK to wear masks. Of course, in a pandemic situation, it makes a lot of sense to ask, "is there a duty to wear a mask?” We would answer that question from the perspective of communitarianism: "In a pandemic, it is natural that we have a moral obligation (for the public good) to wear a mask in order not to spread it to others.”

I think that most people will be non-plussed by the idea that there is a right to wear a mask since, at one end of the scale people generally have a right to wear. There are of course a few instances when this has been challenged (e.g., the veil or hijab etc.). You don't actually say very much, empirically, about the specific use of surgical face-masks in the context of infectious diseases; evidence that the wearing of masks does convey significant benefit would strengthen your claim that it is based on a 'right to life'.

Surgical face-masks are probably more effective in preventing and spreading infection. However, sufficient evidence is not yet available, so we are not discussing mask types or differences in effectiveness here.

Having made the claim that there is such a right you do not go on to develop the more interesting implication that there is a duty for the state to provide masks It seems to me that without this acknowledgement mask-wearing is just a personal preference rather than a substantive right.

Since there is an obligation behind the right, the obligation to supply masks naturally arises in certain cases. For example, in Japan, masks were in short supply during the pandemic due to hoarding, etc., and some people were unable to obtain them. The government is obligated to distribute masks to such people. In Japan, masks were distributed by the government and were called "Abeno masks" because it was during the term of former Prime Minister Abe at the time.

There are two other areas that remain under-developed. One is your claim at 157-160 that the right not to mask is being threatened in Japan. This needs some further discussion and in particular the claim would benefit from some account of the importance of this 'freedom' especially as you suggest that mask wearing is both self-protecting and protecting of others, thus engaging the harm principle.

This may be explained by the fact that in Japan, there is a cultural aspect of a lack of resistance to masks. In Japan, cedar pollinosis (such as "hay fever") is prevalent in spring. Masks are an effective means of preventing hay fever, and Japanese people are accustomed to wearing masks. Mask wearing is also strictly enforced at schools. In particular, it is traditionally normal for children who are in charge of serving school lunches to wear masks.

The second area is linked and relates to the culture of individualism, which you note is a particular feature of the USA. I am curious as to what you make of this culture in terms of the implied (robust) right to non-interference. Is this a right that trumps others, is it a right to 'stupidity' or what?

I don't think non-interference is everything. In normal times, this is a right that trumps others; in pandemic times, is it not a right to 'stupidity' not to mask?

I think that you need to do a little more to show that the right to wear a mask is a substantive right (right to life) and not merely a matter of individual preference. However, I still think that the more interesting question is whether there is a duty to wear a mask.

Yes, I would respond that, as mentioned above, in a pandemic there is a moral obligation (for the public good) to wear a mask to avoid spreading the infection to others.

English proof reading was done. Once again, thank you for reading our paper in detail.

Reviewer 2 Report

The communication discusses the “right to mask for self-protection”.

The topic is important.

Given that some controversy and debates have been discussed and that some individuals that were against the protection policies by the government argued about a “right to mask for self-protection”, but actually were against adhering to wear a mask oneself, the authors should explicitly make clear in the abstract and throughout the manuscript that they are not encouraging people to disregard wearing masks if recommended by the governments.

Author Response

Rev. 2 Comments and Suggestions for Authors

The communication discusses the "right to mask for self-protection".

The topic is important.

Given that some controversy and debates have been discussed and that some individuals that were against the protection policies by the government argued about a "right to mask for self-protection", but were actually against adhering to wearing a mask oneself, the authors should explicitly make clear in the abstract and throughout the manuscript that they are not encouraging people to disregard wearing masks if recommended by the government.

Yes, I clearly added the following. However, I could not find appropriate references, so if you would like me to ask for literature citations, I would appreciate if you would let us know the specifics.

There was some controversy and debate, with some opponents of government protection policies arguing for a "right to mask for self-protection" when they were actually against wearing masks themselves. It should also be noted that the government’s encouragement to wear masks does not mean that people are encouraged to ignore them.

English proof reading was done. Once again, thank you for reading our paper in detail.

Round 2

Reviewer 1 Report

Thank you for the response and comments. I still think you could have added some of these to the MS.